# Treatment of Patients with Malignant Peritoneal Mesothelioma

**DOI:** 10.3390/jcm11071891

**Published:** 2022-03-29

**Authors:** Claire Y. Li, Timothy Kennedy, Henry Richard Alexander

**Affiliations:** 1Department of Surgery, New York-Presbyterian Hospital and Weill Cornell Medicine, New York, NY 10065, USA; yul9025@nyp.org; 2Division of Surgical Oncology, Department of Surgery, The Rutgers Cancer Institute of New Jersey, Rutgers Robert Wood Johnson Medical School, New Brunswick, NJ 08901, USA; tk431@cinj.rutgers.edu

**Keywords:** peritoneal carcinomatosis, gastrointestinal cancer, cytoreductive surgery, HIPEC, immunotherapy, PRODIGE, peritoneal mesothelioma

## Abstract

Malignant peritoneal mesothelioma is a rare malignancy arising from the serosa of the peritoneal cavity. It is diagnosed based on suspicious findings on cross sectional imaging and a tissue biopsy showing confirmatory histologic and immunohistochemical features. The disease is hallmarked by its propensity to progress mainly in the peritoneal cavity. In selected patients, surgical cytoreduction and hyperthermic intra-operative peritoneal chemotherapy has become the initial preferred treatment and is associated with provide prolonged in many patients. Systemic chemotherapy using a couplet of cisplatin or gemcitabine with pemetrexed has modest response rates and duration of response. Expression of PD-L1 has been demonstrated in peritoneal mesothelioma tumors and there has been significant interest in the use of check point blockade targeted against PD-L1 in this clinical setting. Future clinical research using a combination of check point blockade with surgical cytoreduction is a high clinical priority.

## 1. Introduction

Malignant peritoneal mesothelioma (MPM) is a neoplastic process that arises from the serosal membranes lining the peritoneal cavity. The most common site for malignant mesothelioma is the pleural cavity and MPM constitutes almost 20% of all mesothelioma diagnoses. This translates to approximately 600–800 new cases in the US annually [1]. In 1908, Miller and Wynn published the first documented case of MPM [2]. In that report, a 32-year-old male with abdominal pain and ascites was noted to have an extensive and diffuse intraperitoneal neoplastic process at surgical exploration that was eventually confirmed as peritoneal mesothelioma. Fifty years later, a review of the literature revealed only 13 pathologically confirmed cases of MPM [3]. However, after that detailed description of the tumor’s pathological features and the identification of asbestos exposure as an etiologic factor in the development of mesothelioma by Selikoff et al., in 1964, there was a marked increase in the number of documented cases reported in the literature over the next decade [4,5]. The first report of a multimodal therapy specifically to treat MPM was a retrospective review of 23 patients treated with adriamycin-based chemotherapy published in 1983 [6].

In contrast to pleural mesothelioma which has a strong predilection to arise in males, the gender distribution of MPM is approximately equal [7,8]. The disease presents from the second to the seventh decades of life and the mean age at presentation is approximately 53 years. Although the association between asbestos exposure and MPM is not as strongly established as it is for the pleural variant, Selikoff et al. followed almost 18,000 asbestos insulation workers for up to fifty years and observed a markedly higher than expected incidence of MPM with a latency period of several decades [9]. One of the striking features of MPM is the significant heterogeneity in its biological behavior. In some patients, the disease will be progressive and refractory to any therapeutic intervention, whereas other MPM patients can live for decades after initial diagnosis. Morbidity and mortality from MPM are almost invariably due to regional peritoneal disease progression, suggesting that therapeutic modalities that can control the process in the abdomen will result in improved survival. Recently, it has been shown that the tumor suppressor gene BRCA associated protein 1 (BAP-1) is mutated in approximately 60–70% of MPM tumor samples [10]. In addition, in approximately 10% of patients, MPM has been shown to be a part of an inherited familial cancer syndrome in which the patient harbors a germline BAP 1 mutation [11].

## 2. Initial Diagnosis and Evaluation

Although MPM is a rare malignancy, it should be included in the differential diagnosis whenever imaging studies demonstrate a diffuse abdominal process that is consistent with malignancy. Radiographically, the disease has several different manifestations. In many patients, it presents with diffuse ascites and mild abdominal pain with associated omental or peritoneal masses [6,12,13] (Figure 1 left panels). Less frequently, the disease will manifest as a diffuse solid process in the abdominal cavity with disease along the visceral mesenteric membranes with evidence of partial small bowel obstruction (Figure 1 right panels). In the latter situation, the disease is not likely to be amenable to surgical resection [14]. Serum cancer antigen (CA)-125 is often elevated; however, this marker alone is not specific and is best used to monitor for disease recurrence or progression [15,16].

A definitive diagnosis is made based on histopathologic criteria from tissue typically obtained under image guided biopsy. Based on NIH consensus guidelines, positive immunohistochemical staining for two or more markers should be present to establish a definitive diagnosis [17]. The antibodies that stain positive in MPM and are most commonly used include calretinin, cytokeratin 5/6, and Wilms Tumor 1 (WT-1), while those that stain negative include CEA, B 72.3, MOC-31, and Ber-EP4 [12,18]. There are three histologic subtypes of MPM; the epithelioid subtype constitutes approximately 70 to 80% of all MPM diagnoses and is associated with a less aggressive biological behavior than the other histologic types. Sarcomatoid MPM and biphasic MPM (both epithelioid and sarcomatoid features) represent the remaining cases and have a more aggressive tumor biology. Finally, tubulopapillary mesothelioma with invasion, a variant of epithelioid MPM, also has a favorable biological behavior.

## 3. Surgical Treatment of MPM

Over the past 30 years, a number of retrospective single center or multi-institutional publications have reported long-term progression-free and overall survival in select patients undergoing cytoreductive surgery (CRS) with heated intraperitoneal chemotherapy (HIPEC) (Table 1). These studies show that, in general, median overall survival is between 3 to 5 years after CRS and HIPEC and 5-year actuarial overall survival is between 36% and 53%.

Yan et al. published the first large series of 405 MPM patients treated with CRS and HIPEC between 1989 and 2009; the reported median overall survival in this series was 53 months, and 5-year survival was 47% [19]. These authors found that epithelioid histology, absence of lymph node metastasis, complete cytoreduction (CC-0 or CC-1), and HIPEC were associated with improved survival. Two publications analyzed the United States NCDB and the French RENAPE database [22,24]. These studies showed that CRS is being increasingly used over time and that low PCI and completeness of cytoreduction were important factors associated with optimal outcome.

More recently, Bijelic et al. published survival data of MPM patients in the National Cancer Database (NCDB) diagnosed between 2003 and 2014 [7]. In this study, 701 patients were treated with surgery while 1055 were not; overall survival was significantly higher in those who underwent surgery compared to not, 38 versus 7 months, respectively. Independent factors found on multivariate analysis to be associated with improved survival included increasing age, female sex, absence of comorbidities, epithelioid histology, surgery, and intra-operative chemotherapy. Interestingly, amongst surgical patients in this series who received chemotherapy, intra-operative chemotherapy was associated with a survival benefit (median survival 66 months) whereas neoadjuvant chemotherapy (median survival 29 months) or adjuvant chemotherapy given post-operatively (35 months) was not. Two studies reviewed the United States SEER database and showed that the use of surgery and some form of perioperative regional chemotherapy treatment were associated with optimal survival [8,21].

A review of the NSQIP database of patients undergoing cytoreduction and HIPEC for a number of different indications reflects the major nature of the operative procedure. In general, the mean duration of the procedure is greater than 8 h and the length of hospitalization is approximately 13 days [25]. The operative mortality associated with cytoreduction and HIPEC is approximately 2% and major morbidity occurs in 20% of individuals. Therefore, patients being considered for cytoreduction and HIPEC should not have comorbid medical conditions that make them high-risk surgical patients. Based on radiographic assessment, the patient should have a high likelihood of having a complete or near complete cytoreduction. A number of factors have been shown in one or more studies to be associated with favorable outcomes including female gender, age less than 60 years, epithelioid histology, optimal cytoreduction, and lower PCI (Table 1). Because male gender and advanced age are independent prognostic factors associated with poor outcome after cytoreduction and HIPEC, males over the age of 60 represent a high-risk group for which initial non-operative therapy should be considered [13,26]. Similarly, patients who have biphasic or sarcomatoid MPM are at risk for rapid recurrence progression and death even after initial successful cytoreduction and should also be considered for initial non-operative therapy. In a report of 108 patients with MPM, tumor staining for Ki-67 of greater than 10% was a potent independent factor associated with shortened progression-free and overall survival [15]. Recently, the presence of baseline thrombocytosis has been found to be an independent prognostic variable associated with aggressive tumor biology [23]. In a review of 100 patients with MPM, approximately 30% were noted to have an elevated platelet count at initial diagnosis. After cytoreduction and HIPEC, median actuarial overall survival was only 12 months in the thrombocytosis group compared to 55 months in patients who had normal baseline platelet counts.

In surgically resectable patients, the goal of CRS is to remove all visible disease in order to achieve either a CC-0 (no visible disease) or CC-1 (<2.4 mm disease) resection [19]. To achieve a complete cytoreduction, peritonectomy along with resection of major visceral organs may be required. One series of 60 patients suggested a survival benefit with complete peritonectomy compared to selective or partial peritonectomy [27]. Additionally, the resection of major organs has not been found to impact major complications or overall survival in a retrospective cohort study and should be considered in order to achieve a complete cytoreduction [28].

In patients with peritoneal recurrence, repeat CRS and HIPEC is a safe treatment modality that offers survival advantages [29,30]. In a study of 377 patients treated with CRS and HIPEC at a single institution, there were no differences in recurrence rate, disease-free survival or overall survival at a median follow-up of 24 months between patients who underwent single versus iterative CRS and HIPEC for MPM [29]. While 90-day major morbidity and 90-day mortality were similar between the two groups, the iterative CRS and HIPEC group had significant higher rates of late complications, 40% versus 18%, respectively. Therefore, patient selection criteria for iterative CRS and HIPEC warrants additional investigation given the possibility of increased late morbidity. Lastly, given the complex and rare nature of this disease, it is not surprising that patients treated at specialized, academic centers had higher 5-year survival compared to those treated at community hospitals based on a recent retrospective study of MPM patients from 2004 to 2016 [31].

Following the surgical resection of peritoneal metastases for any indication, the abdomen is then prepared for HIPEC in order to address the presence of micrometastatic or minimal residual disease in the peritoneum. In almost all published reports on MPM patients, cytoreduction has usually been performed in association with HIPEC, and there are no prospective studies that have addressed the therapeutic contribution specifically of HIPEC in this clinical setting. Two retrospective studies have shown an association with the use of cisplatin and improved survival compared to the use of mitomycin during HIPEC in MPM patients [20,32]. However, because of the largely uncontrolled nature of the studies it is impossible to know whether the choice of chemotherapy or some other factor was responsible for the difference in outcomes between the groups. However, in a large multicenter study reported from the United States it is notable that the salutary effect of cisplatin compared to mitomycin was observed exclusively in patients undergoing optimal cytoreduction and there was no difference in survival between those same groups in patients who are undergoing suboptimal cytoreduction [20] (Figure 2). Recently, data extracted from the RENAPE database showed improved overall survival with combined chemotherapeutic agents versus single-agent HIPEC, while there was no significant difference in overall survival when comparing across agents [33]. However, this study, too, is retrospective and needs to be confirmed with randomized controlled trials. Currently, other forms of intraperitoneal chemotherapy are being investigated as either bridges to surgery in surgically unresectable patients or as adjuncts to HIPEC after CRS. In the case of the former, one recent cohort study described the outcomes of 20 surgically unresectable patients with MPM who received pressurized intraperitoneal aerosol chemotherapy (PIPAC) through standard laparoscopy along with systemic chemotherapy; of these patients, half of them subsequently underwent CRS and HIPEC, suggesting that PIPAC and chemotherapy may serve as neoadjuvant therapy for initially unresectable patients to facilitate eventual CRS [34].

## 4. Systemic Therapy

In 2004, the U.S. Food and Drug Administration (Rockville, Maryland) approved the combination of cisplatin and pemetrexed as first line medical therapy for patients with MPM and in 2008 carboplatin was approved as an alternative to cisplatin. For patients who cannot receive pemetrexed, gemcitabine can be used in combination with cisplatin. These approvals were based on the results of industry-sponsored single arm prospective clinical trials that established response rates and the toxicity profiles of the regimens. In 2005, Jänne et al. reported the results of pemetrexed alone or in combination with cisplatin for 98 patients with MPM who were deemed surgically unresectable [35]. The response rates for chemotherapy-naïve patients versus those who had previously received chemotherapy were similar (25% and 23.3%, respectively), and the median survival of patients was 13.1 months. The rate of disease control (responding or stable disease) among all patients was 71%. Because of this study, pemetrexed with cisplatin was widely adopted as the preferred initial chemotherapeutic regimen for MPM patients with surgically unresectable disease. The results of a second phase II trial evaluating the efficacy and toxicity profile of pemetrexed and gemcitabine in chemotherapy-naïve patients were published in 2008 [36]. The median overall survival of all patients was 27 months with an estimated one-year survival rate of 68%. The median time to disease progression was 10.4 months and the rate of disease control was 67%. Unfortunately, the toxicity associated with this regimen was significant; 25% of patients did not finish the planned course of therapy.

More recently, there has been significant interest in the use of immune checkpoint inhibitors either alone or in combination with chemotherapy for patients with mesothelioma. Most of the studies that have been conducted over the last 5 years have largely focused on patients with pleural mesothelioma and the exact relevance of these findings to patients with MPM is not exactly known. The PROMISE-meso trial was the first randomized controlled trial investigating the efficacy of anti-PD1 therapy in relapsed pleural mesothelioma; in this multicenter phase III study, 144 patients were randomized to either pembrolizumab or single-agent chemotherapy [37]. The objective response rate was significantly higher in the pembrolizumab arm; however, there was no significant difference found in progression-free survival or overall survival between the two agents. Following the PROMISE-meso study, another important randomized controlled evaluating immunotherapy in pleural mesothelioma was the Checkmate-743 study which compared dual checkpoint blockade (CBP) versus chemotherapy as first-line treatment in patients with pleural mesothelioma [38]. The study randomized over 600 patients and demonstrated a statistically significant and clinically meaningful benefit for patients treated with a combination of nivolumab and ipilimumab compared to systemic couplet chemotherapy (cisplatin and pemetrexed). The actuarial 2-year survival was 41% in the dual CPB group versus 27% in the chemotherapy group. Notably, the benefit of dual CPB was most pronounced in the subset of patients who had high-grade non-epithelioid mesothelioma. Based on the results of the study, the U.S. Food and Drug Administration approved dual CPB in October 2020.

There are very limited data on the use of immunotherapy in patients with MPM. A phase II trial investigated the safety and efficacy of combined bevacizumab and anti-PD-L1 monoclonal antibody atezolizumab in 20 patients with advanced and unresectable MPM [39]. The investigators reported an objective response rate of 40%, with progression-free and overall survival at one year of 61% and 85%, respectively. The JAVELIN study was a single-arm prospective clinical trial evaluating the efficacy and toxicity profile of avelumab, an anti-PDL-1 monoclonal antibody, in 53 patients with predominantly pleural and peritoneal mesothelioma [40]. The study showed a low overall partial response rate of approximately 10%. However, the duration of those responses was quite durable (15 months). Notably, in the subset of patients that had high PD-L1 expression in tumor (>5%), the overall response rate was significantly higher. Recently, the results of the CONFIRM trial demonstrated superior progression-free survival and overall survival in patients with relapsed pleural or peritoneal mesothelioma who were randomized to anti-PD-1 therapy compared to placebo [41]. These data are provocative as there is emerging evidence that PD-L1 tumor expression may be expressed in a large proportion of patients with MPM [42]. In addition to investigations into CPB for the treatment of MPM, there is currently an ongoing phase II clinical trial investigating the feasibility and result of using adjuvant dendritic cell-based immunotherapy after CRS and HIPEC for epithelioid MPM [43].

## 5. Conclusions

MPM is a rare malignancy; etiologically, patients with germline BAP 1 mutations and patients with history of asbestos exposure are at increased risk of developing the disease. Its predilection for progression within the peritoneal cavity has served as the foundation for the use of aggressive local regional treatment strategies, most notably cytoreduction with HIPEC. In select patients, the use of cytoreduction with HIPEC is associated with durable progression-free and overall survival. Systemic therapies are typically reserved for patients who are not candidates for surgical treatment or in patients with high risk of early recurrence after cytoreduction. Going forward, the combination of checkpoint blockade with surgical resection or the use of regional intra-cavitary immunotherapy deserves continued investigation [44].

## Figures and Tables

**Figure 1 jcm-11-01891-f001:**
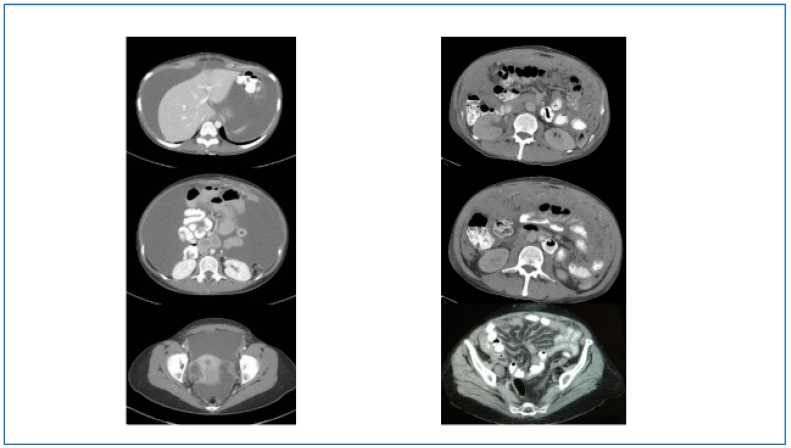
(**Left Panels**): Computed tomography scan of an MPM patient showing typical findings of diffuse ascites and a subtle omental mass. Note the lack of nodularity along the peritoneal surfaces. (**Right Panels**): Computed tomography scan of a patient with diffuse infiltrative MPM distributed extensively along the small bowel mesentery. This type of radiographic picture usually indicates that cytoreduction will not be successful.

**Figure 2 jcm-11-01891-f002:**
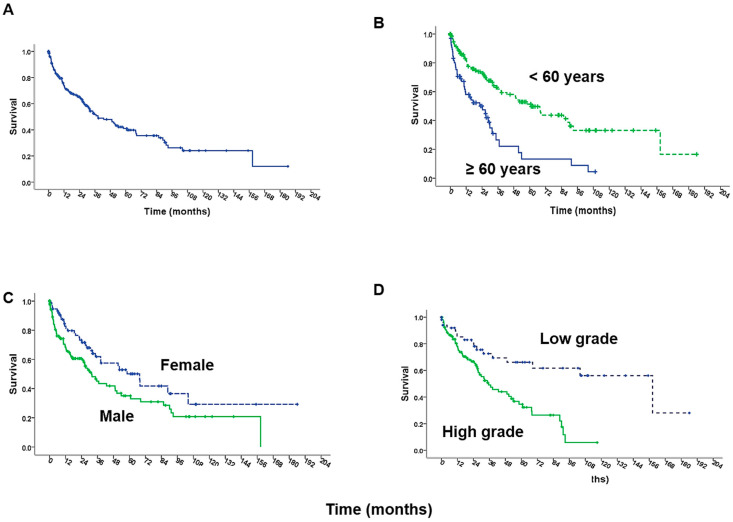
Actuarial overall survival in 211 MPM patients after CRS and HIPEC (**A**) and based on age (**B**), gender (**C**), and histologic grade (**D**). (Reprinted with permission from Alexander et al. [32]. Copyright 2013 Mosby, Inc.).

**Table 1 jcm-11-01891-t001:** Selected series of outcomes in MPM patients.

Study	N	Median OS (Months)	5-y OS	Favorable Prognostic Factors
Yan 2009 [19] Multi-center international	405	53	47%	Epithelioid histology Negative LNs, Optimal CCR Use of HIPEC
Schaub 2012 [16] Single institution	104	N/A	46%	Low PCI Histologic grade Low pre-op CA-125
Baratti 2013 [15] Single institution	108	63	N/A	Low Mitotic count (Ki-67) Epithelioid histology, Optimal CCR
Alexander 2013 [20] Multi-center U.S.	211	38	41%	Histologic grade, Optimal CCR Age < 60 years Use of Cisplatin Female gender
Helm 2014 [21] Miura 2014 [8] SEER database	1047 1591	N/A 38	42% N/A	Use of Surgery Use of Cisplatin Use of EPIC
Kepenekian 2016 [22] RENAPE	126	61	53%	PCI < 30, ASA Score ≤ 2 CCR 0/1 No change OS with neoadjuvant chemo
Li 2017 [23] Single institution	100	33	36%	Thrombocytosis (−) Optimal CCR, PCI ≤ 20
Naffouje 2018 [24] NCDB	1740	52–57	N/A	No change in OS with surgery alone v neoadjuvant v adjuvant chemotherapy. OS: Surgery > chemotherapy > BSC
Bijelic 2020 [7] NCDB	1756	38	N/A	Increasing age, female sex, no comorbidity, epithelioid histology, surgery, chemotherapy

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
