# Peer review of "Treatment of Patients with Malignant Peritoneal Mesothelioma"

_jcm, 2022, doi:10.3390/jcm11071891_

Round 1
Reviewer 1 Report
The authors have very well summarized the most important aspects concerning the treatment of Peritoneal Malignant Mesothelioma.
In view of the interesting chapter concerning the systemic treatment of peritoneal malignant mesothelioma (line 175-234), I would suggest changing the title of this work by removing the word "surgical". "Treatment of Malignant Peritoneal Mesothelioma" seems to be more consistent with the content.
I also suggest the following revisions:
line 62-67: the classification should be revised. Malignant mesothelioma is generally classified into three types: epithelioid (with multiple variants), sarcomatoid (with multiple variants) and biphasic (presence of epithelioid and sarcomatoid cell elements).
line 64: it is incorrect to say that malignant epithelioid mesothelioma is associated with favorable biological behavior as it would be a contradiction in terms. Probably the authors mean that the behavior of epithelioid is less aggressive than the other two tipes (sarcomatoid and biphasic).
line 65-68: the sentence: "Finally, tubulopapillary mesothelioma, a variant of epithelioid MPM, also has a favorable biological behavior" should be revised. it is true that there is a tubular and papillary variant of the malignant form (with infiltration) and therefore it remains an unfavorable biological behavior; probably the authors refer to the Well Differetiated Papillary Peritoneal Mesothelioma (which does not present infiltration) which, however, is not classified in the context of malignant mesotheliomas
Finally, prognostic factors related to HIPEC CRS treatment and which are fundamental in patient selection are mentioned in several points. However, the Authors should also consider an important prognostic factor that is the proliferative index measured with MIB-1 or Ki67 (which was also suggested by the PSOGI guidelines Eur J Surg Oncol. 2020 Feb 28:S0748-7983(20)30114-1) which, when associated with PCI, allows an effective preoperative definition of the treatment program of patients with malignant peritoneal mesothelioma.
Author Response
The authors wish to thank the reviewer for their efforts. The question posed by the reviewer is an excellent one and is the subject of ongoing research at a number of centers around the world. Our laboratory previously published data showing that within a cohort of 40 epithelioid mesothelioma samples, there were distinct genetic signatures associated with shortened survival. The genes that were highly expressed in the poor prognosis subset were largely related to the IL-1 receptor accessory protein pathway. Because these findings have not translated into any clinical application, the information was not included in the current review.
Reviewer 2 Report
We thank the authors for their contribution in the diagnosis, evaluation and management (surgical and medical approach) in patients with peritoneal carcinosis due to malignant peritoneal mesothelioma.
Question for the authors:
Do they believe there are genetic alterations or variants regardless of the histological type that predict a better response to the treatments and so an higher survival
Author Response
The authors thank this reviewer for a number of very helpful comments. We have addressed each as specified below:
1. The title has been changed as suggested.
2. The description of the histologic subtypes of malignant peritoneal mesothelioma has been changed. We appreciate the reviewer's important comments.
3. We have changed the description of the biological behavior of epithelioid mesothelioma as suggested.
4. We have corrected the description of invasive papillary mesothelioma.
5. We have added a sentence describing the data that identify Ki-67 as an important independent prognostic factor in this clinical setting. Importantly, we identified an error in the bibliography that had an incorrect reference. We have corrected the reference which sites the original paper that identified Ki-67 as a prognostic factor. Fortunately, this did not require any other changes or reorganization of the reference list.